# Rich Structural Index for Stereoscopic Image Quality Assessment

**DOI:** 10.3390/s22020499

**Published:** 2022-01-10

**Authors:** Hua Zhang, Xinwen Hu, Ruoyun Gou, Lingjun Zhang, Bolun Zheng, Zhuonan Shen

**Affiliations:** 1School of Computer Science and Technology, Hangzhou Dianzi University, Hangzhou 310018, China; zhangh@hdu.edu.cn (H.Z.); huxinwen@hdu.edu.cn (X.H.); grygry@hdu.edu.cn (R.G.); blzheng@hdu.edu.cn (B.Z.); shenzhuonan@hdu.edu.cn (Z.S.); 2Key Laboratory of Network Multimedia Technology of Zhejiang Province, Zhejiang University, Hangzhou 310018, China; 3Key Laboratory of Brain Machine Collaborative Intelligence of Zhejiang Province, Hangzhou Dianzi University, Hangzhou 310018, China

**Keywords:** depth information, image pyramid, cyclopean map, structural index, visual sensitivity

## Abstract

The human visual system (HVS), affected by viewing distance when perceiving the stereo image information, is of great significance to study of stereoscopic image quality assessment. Many methods of stereoscopic image quality assessment do not have comprehensive consideration for human visual perception characteristics. In accordance with this, we propose a Rich Structural Index (RSI) for Stereoscopic Image objective Quality Assessment (SIQA) method based on multi-scale perception characteristics. To begin with, we put the stereo pair into the image pyramid based on Contrast Sensitivity Function (CSF) to obtain sensitive images of different resolution. Then, we obtain local Luminance and Structural Index (LSI) in a locally adaptive manner on gradient maps which consider the luminance masking and contrast masking. At the same time we use Singular Value Decomposition (SVD) to obtain the Sharpness and Intrinsic Structural Index (SISI) to effectively capture the changes introduced in the image (due to distortion). Meanwhile, considering the disparity edge structures, we use gradient cross-mapping algorithm to obtain Depth Texture Structural Index (DTSI). After that, we apply the standard deviation method for the above results to obtain contrast index of reference and distortion components. Finally, for the loss caused by the randomness of the parameters, we use Support Vector Machine Regression based on Genetic Algorithm (GA-SVR) training to obtain the final quality score. We conducted a comprehensive evaluation with state-of-the-art methods on four open databases. The experimental results show that the proposed method has stable performance and strong competitive advantage.

## 1. Introduction

With the development of three dimensional (3D) technology, 3D visual quality assessment has been an increasingly significant problem in multimedia information processing and communication systems [1]. However, due to the limitation of equipment conditions, stereo images produce various types of distortion in the process of acquisition, storage, encoding, transmission and compression, etc., which cause the image quality to decrease. Poor quality images cause poor subjective feelings [2,3,4]. Therefore, it is necessary to establish an effective method to measure the quality of stereo images. Since human beings are the ultimate receivers of sensory information in most real applications. In consequence, it is considered as the most accurate way to evaluate the image quality that human beings score images according to their subjective perception. This method is called subjective quality assessment. However, subjective quality assessment suffers from cost and time issues, and it is easily interfered by the subjective consciousness of evaluators and external environment. These defects lead to a substantial decline in the usefulness of subjective evaluation. As a result, the objective quality evaluation algorithm is worth studying. Due to the physiological structure of the eyeball is very complex, compared with the absolute authenticity of subjective quality scores, the objective quality evaluation model can not completely simulate the HVS. In order to strive to maintain consistency with human perception, rich image structural features play particularly important in the area of image assessment. HVS is very sensitive to image structure characteristics in perceiving distortions. Rich structural indices can be used as important features to measure the stereo image quality. The multi-scale structure similarity of image pair produces much better results than its single-scale structure counterpart in [5]. The study [6] used the gradient similarity to measure the change in contrast and structure in images. Moreover, the research [7] used SVD combing with local structure to better measure the sharpness of the picture.

However, in the field of SIQA, simple structure index is not easy to design an accurate SIQA metric owing to the disparity and depth. The study [8] found that the absolute disparity map approximately reflects the disparity and depth.The research [9] proposed that the stereoscopic images should classify into non-corresponding, binocular fusion, and binocular suppression regions.

As aforementioned, combining the importance of rich image structure to HVS and the theory of binocular, we propose a RSI-SIQA model based on multi-scale visual perception characteristics. Based on the internal hierarchy, depth structure, luminance texture of stereo images, our model calculates three contrast indices, and then uses the GA-SVR [10] model to predict the objective quality scores of stereo pairs. The main contributions of our work are summarized as follows:Considering the image edge texture structure and internal hierarchical structure, we propose local Luminance and Structural Index (LSI) and the Sharpness and Intrinsic Structural Index (SISI) introducing image pyramid and cyclopean map to express the binocular perception characteristics of image information at different viewing distances;Binocular parallax is the most important physiological basis for human beings, which can reflect depth perception information. Towards this end, we advance Depth Texture Structural Index (DTSI) which combines the disparity map and the cross-mapping of gradient with sensitive factors to build a model extracting depth information closer to human visual subjective perception.

The rest of the paper is organized as follows. In Section 2, we introduce related works about cyclopean map and SIQA. In Section 3, we introduce the proposed RSI-SIQA method. In Section 4, we conducted experimental comparisons and analyzed performance on four public stereo databases. Finally, conclusions are drawn in Section 5.

## 2. Related Work

After studying the physiological structures of HVS, some scholars put forward the concept of cyclopean maps. An image combined by left and right views through weight factors is called cyclopean map. Maalouf et al. [11] first proposed the cyclopean map paradigm. They first build cyclopean map on original and distorted stereo pairs, then calculated the sensitivity coefficients of cyclopean maps. Finally, they calculated the final quality score by comparing the correlations between these coefficients. Chen et al. [12] proposed a Full Reference Stereoscopic Image Quality Assessment (FRSIQA) method based on binocular competition. They applied the MS-SSIM [5] index to original and distorted cyclopean maps to obtain the objective scores. In addition, they used Gabor filters to calculate the convex weighting factors for left and right eyes. Fezza et al. [13] proposed a method similar to Chen, except that they used local entropy as a weight factor to combine left and right views. Besides, they also applied the MS-SSIM index to the disparity map of distorted images. The performance of this method model is significantly improved compared to Chen. Lin et al. [14] proposed a FRSIQA algorithm based on cyclopean amplitude map and cyclopean phase map. They applied visual saliency to the correction of the amplitude and used a spatial-domain binocular modulation function to predict the final quality score. Yang et al. [15] proposed a FRSIQA method based on the saliency of the cyclopean maps. They first build the cyclopean maps of reference and distorted stereo pairs, then calculated corresponding saliency maps, and finally compared the features of saliency maps to obtain a quality score. Cyclopean maps effectively improve the prediction performance of a SIQA model, which is also explained in the above literature. However, the cyclopean map model with image pyramid has not been studied. Besides, the calculation method of binocular weight factors also significantly affect the performance of cyclopean map. We will study binocular sensitivity in this paper.

In the research of Stereoscopic Image Quality Assessment (SIQA) method, depth information is a very important difference feature. Khan et al. [16] proposed a FRSIQA method based on sparse representation by combining depth information and sparse dictionary. They firstly represented luminance images and depth images sparsely, and then measured the sparsity changes of reference and distortion stereo pairs in luminance domain and depth domain respectively under different constraints to calculate the quality score. Jiang et al. [17] proposed a FRSIQA method based on the 3D visual attention model, which took center deviation and depth information into account, and applied various combination models to calculate the final quality score. Liu et al. [18] proposed a FRSIQA method based on a binocular fusion significance model. They applied the disparity model and the Discrete Cosine Transform (DCT) coefficient model to stereo saliency map, then combined binocular views on this basis. Finally, they extracted the features of saliency cyclopean maps. Yao et al. [19] proposed a FRSIQA algorithm based on visual significance features and gradient edges. They performed gradient processing on the gradient amplitude map and depth map of reference and distorted stereo pairs to obtain the quadratic gradient map and depth gradient map, and then compared visual significant features to obtain the final objective quality. The method of Khan et al. [20] is similar to Yao, except that they considered the saliency region of the depth edge and proposed a new model of Saliency Edge based on Depth perception (SED). Finally, they incorporated monocular features using a geometrically weighted algorithm. Although our operation on disparity maps is similar to Khan method, we combine the image pyramid model with depth information. In addition, we have improved the features extraction method and the combine method, which will be described in detail below.

## 3. Materials and Methods

Many researchers show that the HVS detect image distortions by measuring image structure information (such as edges, textures) [21,22]. Furthermore, it has been widely accepted that viewing distance has great impact on image quality perception [23]. We propose a RSI-SIQA model in this paper by combining LSI, SISI and DTSI. The framework of the proposed method is shown in Figure 1. In the first place, we put the grayscale reference pair and distortion pair which converted from RGB images into IPC. Then, by quantifying the sensitivity of the stereo pair into weight factors and obtainting cyclopean map in gradient domain, we obtain LSI. Similarly, in the SVD domain, the singular value based on the previous method we obtain from stereo pair is SISI. At the same time, we combine the gradient cross-mapping algorithm and hypotenuse gradient on the disparity maps to obtain DTSI. Next we calculate contrast similarity deviation of the indices from original pair and distorted pair. In the end, all the features we obtain are input into the GA-SVR model for training to obtain the final quality score.

### 3.1. Image Pyramid Based on CSF (IPC)

Contrast is the carrier or medium of visual information [24]. Many works [25,26,27] not only use contrast as an attribute to describe image quality but also apply in the field of image enhancement. The contrast is sensitive to the spatiotemporal frequency and viewing distance. The image pyramid composed by different resolutions images can reflect the relationship between images and view distance. At the same time, CSF plays a vitally important role in HVS which has diverse sensitivities to distortions depending on spatial frequency. Based upon above reasons, we used IPC before extract our features to have a better perceptibility and increase the flexibility of SIQA.

#### 3.1.1. Image Pyramid

Image pyramid is the sampling of signals of different granularity. In many cases, multi-scale signals which make up the image pyramid actually contain different features. The sampling method can be non-overlapping or overlapping. If the sampling scale factor is 2, for each additional layer, the row and column resolution is 1/2 of the original. It has been pointed out that the quality of experience is seriously influenced by image scales [23]. In other words, it is also greatly affected by viewing distance. Figure 2 is an example. This example illustrates the relationship between image scales and human visual perception. Image (a) shows the original image and the corresponding distortion components. Image (b), (c), (d) are obtained by down-sampling the image (a) in both horizontal and vertical directions by 2×, 4× and 5× times, respectively. In this work, in order to avoid generating additional distortions, we use imresize function with bicubic interpolation algorithm to conduct the down-sampling operation. Bicubic interpolation is an effective interpolation algorithm that can produce high-fidelity images. We highlighted three border areas in each images with red rectangles. Image (a) is a normal scale view, and we can easily observe the difference between the two views, especially in the boundary region. However, as the scale of image decreases (equivalent to increasing the viewing distance), the difference between original and distorted images is gradually narrowing. In image (d), we can hardly observe any difference. The main reason for this phenomenon is that as the viewing distance increases, the viewing angle decreases and less structural features of image can be noticed. This example shows that in the same natural environment, human visual perception of image details mostly depends on the effective scale of the HVS. Therefore, it can be concluded that humans beings have different visual perception quality for different viewing distances. It is necessary to apply image pyramid to SIQA method.

#### 3.1.2. Contrast Sensitivity Function (CSF)

The experiment in [28] shows that spatial CSF is more sensitivity in low frequency. At the same time, the study [29] also indicates that the sensitivity of HVS is different to distortions which can be reflected by CSF, especially in the boundary region. Thus, in this paper, based on the image pyramid, we apply the adjusted CSF filter to reference and distorted stereo pairs to reflect the binocular sensitivity of visual stimuli. Here, in order to smooth the image, we set the frequency response of the circular symmetric Gaussian filter, $h_{1}\left(f,\theta \right)$, is denoted by:(1)$$h_{1}\left(f,\theta \right)=exp\left(-2\pi^{2}\sigma^{2}f^{2}\right),$$
where *f* denotes the radial spatial frequency in cycles per degree of visual angle (c/deg), θ∈[−π,π] denotes the orientation. The parameter σ is used to control the cutoff frequency of the filter. In this experiment, in order to capture the edge components related to binocular perception while filtering out the high frequency components irrelevant to perception, we set a 3×3 filter window size and σ=2.

Referring to the frequency response of the CSF model initially introduced by Mannos and Sakrison [30] with specifically modified by Daly [31], $h_{2}\left(f,\theta \right)$ is denoted by:(2)$$h_{2}\left(f,\theta \right)=2.6\left(0.0192+\gamma f_{\theta }\right)exp\left[-\gamma f_{\theta }\right],$$
where γ=0.114 [32,33]. $f_{\theta }=\frac{f}{0.15cos(4\theta )+0.85}$, which cause oblique effect. Combining Equations (1) and (2), the final adjusted CSF model $h_{csf}\left(f,\theta \right)$ is given by:(3)$$h_{csf}\left(f,\theta \right)=h_{1}\left(f,\theta \right)h_{2}\left(f,\theta \right).$$

We input the reference and distorted stereo pair into the image pyramid, and then input the stereo pair with different scales into the CSF filter. Finally we obtain the image $CF_{o,v}^{k}$, $CF_{o,v}^{k}$ is denoted by:(4)$$CF_{o,v}^{k}=F^{-1}\left[h_{csf}\left(f,\theta \right)F\left[I_{o,v}^{k}\right]\right]$$
where F[·] and $F^{-1}\left[\cdot \right]$ denote the Discrete Fourier Transform (DFT) and inverse DFT respectively, o∈{r,d}, v∈{L,R}, *r* and *d* represent reference and distortion components, *L* and *R* represent the left and right stereo image, respectively. k=0,1,…,n, *k* represents the number of iterations of the image pyramid acting on luminance images. Note that the original scale image also contains some non-negligible visual information. So, we must also perform the same processing on the original scale stereo pair, k=0 denotes the original scale size.

### 3.2. Rich Structural Indexes (RSI)

#### 3.2.1. Local Luminance and Structural Index (LSI)

Gradient is often used to describe structural feature of images. So as to measure luminance and structure feature more effectively, we adopted weighted gradient to obtain LSI. To clearly state the workflow of the LSI, we give its algorithm description in Figure 3.

Perceptually features at different image scales can be modeled by Gaussian derivative functions in terms of retino-cortical information [34], and can be represented using gradient magnitudes. Gradient have been shown to capture the structural features of images effectively [6,35,36]. Here, we use Sobel operator [37] to process images $CF_{o,v}^{k}$ to obtain gradient graph $GM_{o,v}^{k}$. Considering the effect of luminance masking and contrast/texture masking will influence edge visibility, thus weakening structural information. Therefore, we perform a locally adaptive normalization process on gradient maps, which can enhance the locally edge structure. The locally weighted gradient map $WGM_{o,v}^{k}$ is calculated as follows:(5)$$WGM_{o,v}^{k}=\frac{GM_{o,v}^{k}\left(x,y\right)}{\gamma_{o,v}^{k}\left(x,y\right)+\varepsilon },$$
a small normal number ε can avoid numerical instability when $\gamma_{o,v}^{k}\left(x,y\right)$ has a small value. The weighted window $\gamma_{o,v}^{k}\left(x,y\right)$ is calculated as follows:(6)$$\gamma_{o,v}^{k}\left(x,y\right)=\sqrt{{\sum \sum }_{\left(x^{\prime },y^{\prime }\right)\in \Omega_{x,y}}MF_{o,v}^{k}\left(x^{\prime },y^{\prime }\right)\omega \left(x^{\prime },y^{\prime }\right)}$$
with
(7)$$MF_{o,v}^{k}\left(x,y\right)=\frac{{\left(GM_{o,v}^{k}\right)}_{.}^{\;2}+{\left(CF_{o,v}^{k}\right)}_{.}^{\;2}}{2},$$
where $\Omega_{x,y}$ is the local window centered at (x,y), $\omega (x^{\prime },y^{\prime })$ are positive symmetric weights satisfying $\sum_{x^{\prime },y^{\prime }}\omega \left(x^{\prime },y^{\prime }\right)=1$. “$.^{\;2}$” indicates element-wise square of the matrix. We use the Gaussian kernel with window size 3×3 and σ=2. The locally weighted gradient operation can more clearly reflect the local variation in edge structure characteristics of stereo images caused by different types and levels of distortion. The above mentioned can be shown in Figure 4, the first row is original image of left view. The second row is the normal gradient image. In addition, the third row is the weighted gradient image obtained by our method.

Because the binocular perception characteristics of stimuli are different, the binocular visual perception characteristics need to be further integrated after the gradient processing of monocular views. Here, we use the cyclopean map algorithm. The cyclopean map model can be expressed as:(8)$$WGM_{o}^{k}=W_{o,L}\cdot WGM_{o,L}^{k}+W_{o,R}\cdot WGM_{o,R}^{k},$$
where o∈{r,d}, $W_{o,L}$ and $W_{o,R}$ represent the left and right weighting factors, which indicate the binocular acceptance ratio of information when receiving the same stimulus. The left and right eyes respond different to stimulus, which is the most manifestation of binocular sensitive performance. From the side, it can also reflect the left and right eyes have different perception characteristics of information. It is known that CSF filtering can capture the relatively important structural frequency components of the HVS in a certain spatial frequency range. In this work, we will quantify the frequency component of the original scale image filtered by CSF as the binocular sensitivity. To be specific, $W_{o,L}$ and $W_{o,R}$ can be denoted by:(9)$$W_{o,L}=\frac{MCF_{o,L}}{MCF_{o,L}+MCF_{o,R}}$$
(10)$$W_{o,R}=\frac{MCF_{o,R}}{MCF_{o,L}+MCF_{o,R}},$$
where $MCF_{o,v}$ represents the frequency energy in the low-frequency space of the original scale images after CSF filtering, which also indicates that most of the information perceived by the HVS is concentrated in the low-frequency space. $MCF_{o,v}$ is calculated as follows:(11)$$MCF_{o,v}=\frac{\sum_{x=1}^{M}\sum_{y=1}^{N}CF_{o,v}\left(x,y\right)}{MN},$$
where M×N represents the size of the image, v∈{L,R}. In the end, we obtain the cyclopean maps of reference and distorted stereo pairs at all scales.

#### 3.2.2. The Sharpness and Intrinsic Structural Index (SISI)

In addition to the luminance and texture structure features, the intrinsic structure of images is also a very important visual perception information. Singular value vector can effectively reflect the internal hierarchical structure changes of images [38]. Furthermore, the singular value not only reflects the strength of the gradients along the dominant direction and its vertical direction but also is sensitive to blur, so it can be a sharpness metric [39]. To clearly state the workflow of the SISI, we give its algorithm description in Figure 5.

Same as above, we first put the original pair and the distorted pair to IPC, Then we use SVD on the sensitive pair. For an M×N image *I*, it can be decomposed by:(12)$$I=USV^{T},$$
where *U* and *V* are unitary matrices of size M×M and N×N, respectively. *S* is a non-negative singular matrix on the diagonal of magnitude M×N. The columns of *U* and *V* represent the left and right singular vectors of image *I*, respectively. Therefore, *S* is a multi-level matrix of image *I*, which is also called the SVD. *S* represent the energy in the dominant orientation and its perpendicular direction, respectively.

In this work, the SVD of images $CF_{o,v}^{k}$ are denoted by $SVD_{o,v}^{k}$, k=0,1,…,n. Then, we combine the monocular singular value matrix to obtain the singular value cyclopean map, $SVD_{o}^{k}$, is denoted by:(13)$$SVD_{o}^{k}=W_{o,L}\cdot SVD_{o,L}^{k}+W_{o,R}\cdot SVD_{o,R}^{k},$$

#### 3.2.3. Depth Texture Structural Index (DTSI)

The biggest difference between stereo image quality evaluation and 2D image quality evaluation lies in the perception of 3D features. Depth feature is one of the most perceptual features that can best reflect stereo information. To clearly state the workflow of the SISI, we give its algorithm description in Figure 6.

There is also an inseparable correlation between depth information and image scale. According to [40,41], the depth information of stereo images is often reflected by disparity maps. In this work, we combined the IPC and used the binocular matching algorithm based on SSIM [42] to obtain the disparity map at each image scale. Then, we calculate the gradient maps of disparity maps in horizontal and vertical directions, $DG_{o,v}^{i,l}$ and $DG_{o,v}^{i,p}$, *l* and *p* represent the horizontal and vertical directions, respectively. In order to measure the contrast of the intrinsic disparity edge structures between reference and distortion components, we calculate the internal disparity gradient maps as
(14)$$IDG_{v}^{i}=|DG_{r,v}^{i,l}DG_{d,v}^{i,l}|+|DG_{r,v}^{i,p}DG_{d,v}^{i,p}|,$$
where i=1,…,m, v∈{L,R}. *i* represents the number of iterations of the image pyramid model acting on disparity maps. It should be noted that, we do not use the original scale image for depth information features extraction. According to the hypotenuse theorem, the gradient magnitude at different image scales can be denoted as:(15)$$DG_{o,v}^{i}=\sqrt{{\left(DG_{o,v}^{i,l}\right)}_{.}^{\;2}+{\left(DG_{o,v}^{i,p}\right)}_{.}^{\;2}}.$$

In terms of inter-gradient map and gradient magnitude, we obtain the gradient contrast similarity between reference and the distorted stereo pairs, $D_{v}^{i}$, is given by:(16)$$D_{v}^{i}=\frac{2\cdot IDG_{v}^{i}+\alpha_{3}}{{\left(DG_{r,v}^{i}\right)}_{.}^{\;2}+{\left(DG_{d,v}^{i}\right)}_{.}^{\;2}+\alpha_{3}},$$
where $\alpha_{3}$ is the normal number to ensure numerical stability.

To reflect the contrast changes of image depth information, we perform standard deviation operation on disparity gradient contrast similarity to obtain the depth edge structure features of monocular views, $f_{L}^{i}$ and $f_{R}^{i}$. Synthesizing the monocular features to obtain the final depth features, $f_{D}^{i}$, is denoted by:(17)$$f_{D}^{i}=W_{d,L}\cdot f_{L}^{i}+W_{d,R}\cdot f_{R}^{i},$$
where $W_{d,L}$ and $W_{d,R}$ represent the left and right weight factors of distorted stereo pairs, respectively.

### 3.3. Contrast Similarity Deviation

Numerous research works show that there are many types of contrast [43,44]. For example, Weber contrast, Michelson contrast, RMS (root mean square) contrast, etc. Weber contrast is mainly used to describe character features; Michelson contrast is mainly used to describe the gratings, RMS contrast is mainly used for natural scene stimulation. In this work, we adopted the RMS contrast. Specifically, with the gradient cyclopean maps and singular value cyclopean maps, we calculate the similarity between reference and distortion components at each image scale:(18)$$SG^{k}=\frac{2\cdot WGM_{r}^{k}WGM_{d}^{k}+\alpha_{1}}{{\left(WGM_{r}^{k}\right)}_{.}^{\;2}+{\left(WGM_{d}^{k}\right)}_{.}^{\;2}+\alpha_{1}}$$
(19)$$S^{k}=\frac{2\cdot SVD_{r}^{k}SVD_{d}^{k}+\alpha_{2}}{{\left(SVD_{r}^{k}\right)}_{.}^{\;2}+{\left(SVD_{d}^{k}\right)}_{.}^{\;2}+\alpha_{2}},$$
where $\alpha_{1}$ and $\alpha_{2}$ are normal numbers to ensure numerical stability. Finally, in order to accurately reflect the image contrast changes, we use standard deviation to generate edge structure features, $f_{G}^{k}$, is denoted by:(20)$$f_{G}^{k}=\sqrt{\frac{\sum_{x=1}^{M}\sum_{y=1}^{N}{\left(SG^{k}\left(x,y\right)-MSG^{k}\right)}^{2}}{MN}},$$
where k=0,1,…,n, $MSG^{k}$ can be calculated by averaging the similarity map:(21)$$MSG^{k}=\frac{\sum_{x=1}^{M}\sum_{y=1}^{N}SG^{k}\left(x,y\right)}{MN}.$$

Figure 7 shows the curves of gradient similarity deviation between reference and distortion components at each image scale when n=4. Here, we use 4-level and 2-level gaussian blur images. The higher the level, the more distorted the image. Noted that, in order to visually observe the difference between the two, we enlarged the gradient similarity deviation by 100 times before plotting the curves. It is observed that the gradient similarity deviation values of low distortion image are lower than those of high distortion image. Therefore, the gradient similarity deviation can measure the distortion degree of images.

Similarly, the hierarchical structure features $f_{S}^{k}$ can be obtained by performing the same operation on singular value similarity $S^{k}$.

Figure 8 shows the curves of singular value similarity deviation between reference and distortion components at each image scale when n=4. We also magnify the similarity features of singular values by 100 times. It is observed that the singular value similarity deviation values of low distortion image are mostly lower than those of high distortion image. In particular, the difference is most pronounced in the down-sampled 4× and 8× images. As a result, the singular value similarity deviation value can also effectively measure the distortion degree of images.

### 3.4. Final Quality Assessment

In this work, we set n=4, a total of 10+m perceptual features are extracted for each stereo pair, including 5 multi-scale gradient similarity deviation values, 5 multi-scale singular value similarity deviation values, and *m* multi-scale depth texture structure similar deviation values. We do feature normalization on LSI, SISI and DTSI respectively and then concatenate them as feature fusion. We will explain the value of *m* in the experimental analysis section. To apply the extracted perceptual features to the RSI-SIQA model, we adopt GA-SVR to learn in multiple public databases. SVR has been widely used in image quality evaluation [45,46,47,48,49,50]. However, the two parameters of penalty parameter *c* and the $g_{2}$ in SVR are usually selected subjectively based on personal experience, they are very random, and improper selection will affect performance. So in order to make SVR obtain better prediction ability and generalization ability, we introduce genetic algorithm (GA) to optimize SVR [10]. The main steps of GA-SVR are as follows:1.The penalty factor *c* which reflects the degree of penalty of the algorithm on the sample data beyond the pipeline and $g_{2}$ representing the radial basis function in the SVR are coded to generate the initial population.2.The new population is obtained by random cross selection, single point crossover, and mutation with probability 0.7. Then, we calculate the fitness of new population and select the highest fitness.3.Judge whether the highest fitness satisfies the stopping condition. If so, determine it as the optimal parameter combination and apply it to SVR. If not, return to step 2 and start the calculation again.

We optimized SVR parameter combination through GA, which can prevent performance loss due to the randomness of parameters, and at the same time avoid the local optimization of SVR model.

In specific experiments, we input the subjective evaluation values and the combined perceptual features into the GA-SVR model for training and testing, which 80% of training set is used to train the model, and 20% of testing set is used to predict performance. To eliminate prejudice, the training set and testing set are randomly assigned and do not overlap in each iteration. We set 1000 iterations and use the median of all results as the final quality score for this dataset.

## 4. Experimental Results and Analysis

### 4.1. Experimental Databases

In this work, we evaluate our RSI-SIQA algorithm on four public databases.

The LIVE Phase-I 3D database [51] contains 20 original stereo pairs and 365 symmetrically distorted stereo pairs. It has five different types of distortion, namely White Noise (WN), JPEG2000 (JP2K), JPEG, Gaussian Blur (Gblur) and Fast Fading (FF). Except for the WN distortion image only 45 outside, the other four types of distortion image have 80 pairs. The subjective quality score is represented by Differential Mean Opinion Score (DMOS). The image scale of the LIVE Phase-I 3D database is 360×640.

The LIVE Phase-II 3D database [12] contains 8 original stereo pairs and 360 distorted stereo pairs. The LIVE Phase-II is composed of 120 symmetric and 240 asymmetric distorted stereo pairs. Like the LIVE Phase-1 3D database, the LIVE Phase-2 3D database has five types of distortion, except that each distortion type has 72 pairs of images. DMOS and image scale are also the same.

The WaterlooIVC Phase-I 3D database [52] includes 6 original stereo pairs and 324 distorted stereo pairs. The types of distortion are Gaussian White Noise (WN), Gaussian Blur (Gblur), and JPEG compression (JPEG) and each distortion type has four distortion levels. The WaterlooIVC Phase-I consists of 252 asymmetric distortion stereo pairs and 72 symmetric stereo pairs. The image scale of the WaterlooIVC Phase-I 3D database is 1920×1080. Mean Opinion Score (MOS) is the subjective score used by the database.

The MCL 3D database [53] includes 9 original stereo pairs and 648 symmetrically distorted stereo pairs. The types of distortion are additive Gaussian White Noise (WN), Gaussian Blur (Gblur), JPEG, JP2K, Sampling Blur (Sblur) and Transmission Loss (Tloss). The MCL 3D database has 108 pairs of images per distortion type. In this database, 30% of the images have a scale of 1024×728 and the remaining 70% have a scale of 1920×1080. MOS is the subjective score used by the database.

### 4.2. Overall Performance Comparison

We select three commonly used evaluation indicators to evaluate the performance of the SIQA model: Pearson Linear Correlation Coefficient (PLCC) can judge the accuracy of the model; Spearman Rank Order Correlation Coefficient (SROCC), which can judge the monotonicity of the model; Rooted Mean Square Error (RMSE) can reflect the consistency of the model. The PLCC and SROCC are closer to 1, and the RMSE is closer to 0, indicating the prediction performance of the model is better.

In our model, both the cyclopean map model and the depth texture feature are applied. Therefore, in order to verify their impact on the SIQA model, we compared the performance of the method with and without the cyclopean map model on four databases, as well as the performance of the model after adding depth features. On two LIVE databases, we set m=3, and on WaterIooIVC Phase-I and MCL databases, we set m=4. The comparison results are shown in Table 1. $Q_{1}$ represents the performance of measuring LSI and SISI, and $Q_{2}$ represents the performance of measuring only DTSI, respectively, and *Q* denotes the performance combining LSI, SISI and DTSI. Noted that, in the case of not using the cyclopean map, we directly calculate the contrast similarity deviation feature values of monocular views, and then use the binocular sensitivity factors of distorted stereo pairs to synthesize the left and right features. As we can see, most of them show performance degradation compared to *Q*. It indicates that the cyclopean map and depth texture information can improve the predictive performance of the SIQA model.

Table 2 shows the overall performance comparison between the proposed method and six state-of-the-art FR algorithms on two LIVE 3D databases. The best performance results shown in bold, ‘-’ indicates that the value is not available (this partial result is not provided in the original article). Here, we select the methods of Khan [20], Ma [54], Yue [45], Geng [55], Jiang [56], and Shao [57]. It is observed that the overall performance of the proposed method is generally better than that of the comparative methods, especially on the LIVE Phase-II database. Since the Phase-II database contains 66% asymmetric distortion stereo pairs, which conforms to the binocular asymmetric receiving information mechanism. As a result, the experimental measurement on the Phase-II database is more practical.

In order to further prove that the proposed method has a better prediction performance in asymmetric distortion, we select the WaterlooIVC Phase-I 3D database with an asymmetric distortion ratio of 76% for comparative experiments. In this database, we choose five comparison methods including Khan [20], Ma [54], Yue [45], Geng [55] and Yang [49]. The results are shown in Table 3. We can find that the performance of the proposed method is the best in PLCC and SROCC indicators, while the performance in RMSE indicators is a little worse, but is not far from the best. Therefore, it can be concluded that the method has better performance in the prediction of asymmetric distortion.

To illustrate the performance of the proposed SIQA method on different scale images, we selected MCL 3D database containing two image scales for comparison experiments. Here, we choose five comparison methods: Zhou [53], Chen [42], Khan [20], Shao [58] and Liu [59]. The results are shown in Table 4. We can find that the proposed method has obvious competitive advantages by analyzing the data in table. In the MCL database, the number of images is large and the image sizes are inconsistent. Therefore, the prediction performance of the SIQA model on this database is somewhat worse than the previous two databases. Nevertheless, the proposed method still has better performance than the comparative experiments, which indicates that the perceived features extracted by the SIQA model can better reflect the objective quality of stereo pairs.

Furthermore, in order to demonstrate the prediction performance of the proposed method more intuitively, we draw scatter plots between the subjective scores and the predicted objective scores on the four databases, as shown in Figure 9. Because the MCL and WaterIooIVC phase-I databases have MOS fractions, the slope of the scatter plots are opposite to the LIVE databases. The distribution of scatter plots represents the predictive performance of the proposed SIQA method. From this, we can draw a conclusion that the objective value predicted by the proposed SIQA method is highly consistent with the subjective value.

### 4.3. Single Distortion Performance Comparison

In order to measure the performance of the proposed SIQA method more comprehensively, we perform a comparative experiment on a single distortion type stereo pair on LIVE 3D databases and MCL 3D database. Table 5 and Table 6 show the comparison results of the five distortion types on LIVE 3D databases. After analyzing the data in tables, we find that most of the results of the proposed method are better than the comparison methods in PLCC and SROCC indices. Although the performance of the proposed method is slightly worse on WN and FF, it achieves the best performance on JPEG. Table 7 and Table 8 show the comparison results of the six distortion types in MCL 3D database. We clearly observe that our method has obvious advantages in PLCC and SROCC. It can be concluded that the objective values predicted by the proposed method fit well with the subjective values.

## 5. Conclusions

In this paper, we propose a RSI-SIQA model combining image pyramid and visual perception characteristics. Our method considers not only the edge structures and hierarchical structures of the multi-scale cyclopean maps, but also considers multi-scale depth edge structures. According to the experimental results, it can be found that the performance of the proposed model can be improved after adding cyclopean map and depth information. We conducted comparative experiments on four public databases and proved that the proposed method has good stability and high prediction performance. The proposed method in this paper is mainly based on the idea of machine learning. In the next step, we will study the visual characteristics based on the idea of deep learning and design a more effective objective quality evaluation model.

## Figures and Tables

**Figure 1 sensors-22-00499-f001:**
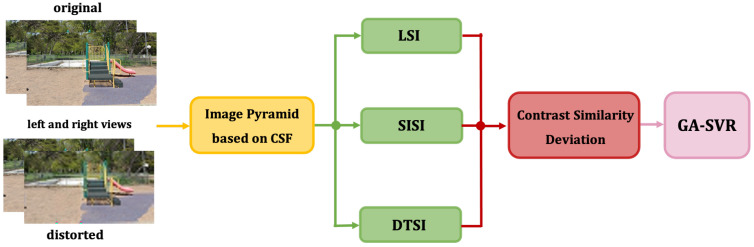
Framework diagram of the proposed method.

**Figure 2 sensors-22-00499-f002:**
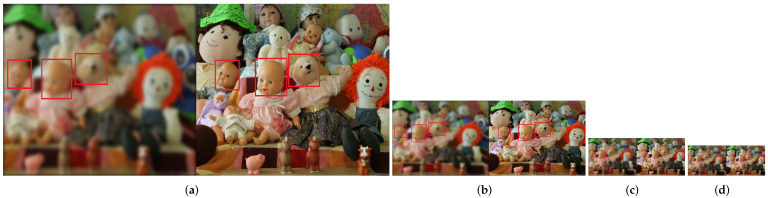
Image pairs with different scales. (**a**) Original image pair. (**b**) down-sampled image pair by 2× times. (**c**) down-sampled image pair by 4× times. (**d**) down-sampled image pair by 5× times. Three representative regions are highlighted with red rectangles in each image.

**Figure 3 sensors-22-00499-f003:**
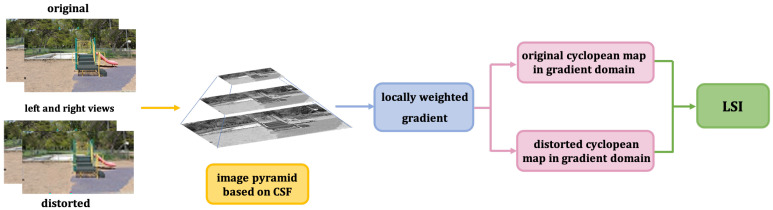
Workflow diagram of LSI.

**Figure 4 sensors-22-00499-f004:**
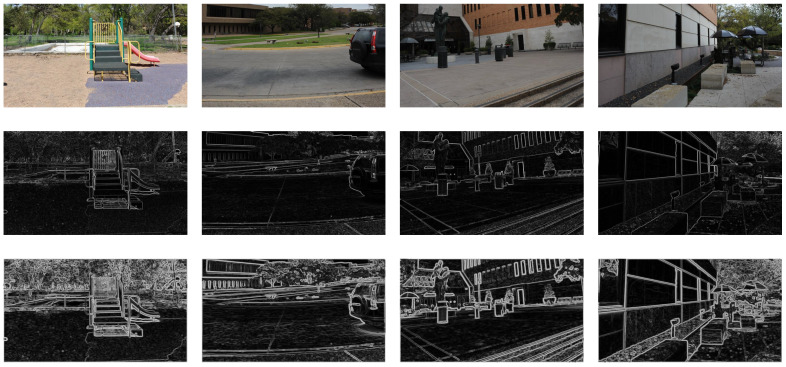
Comparison of two methods of gradient. From (**top**) to (**bottom**), original maps, normal gradient maps and locally weighted gradient maps.

**Figure 5 sensors-22-00499-f005:**
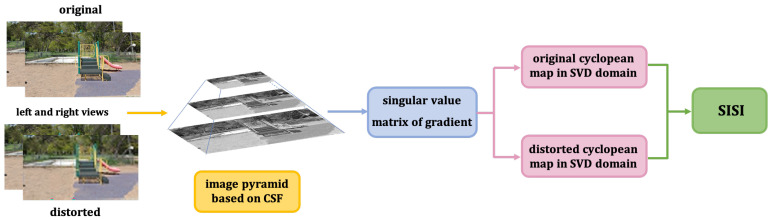
Workflow diagram of SISI.

**Figure 6 sensors-22-00499-f006:**
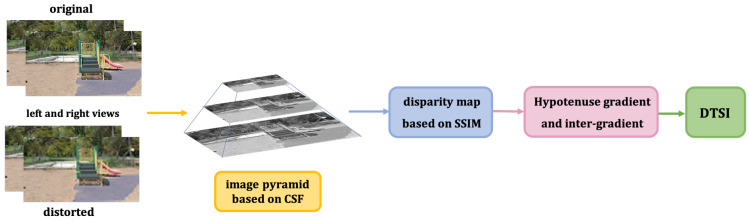
Workflow diagram of DTSI.

**Figure 7 sensors-22-00499-f007:**
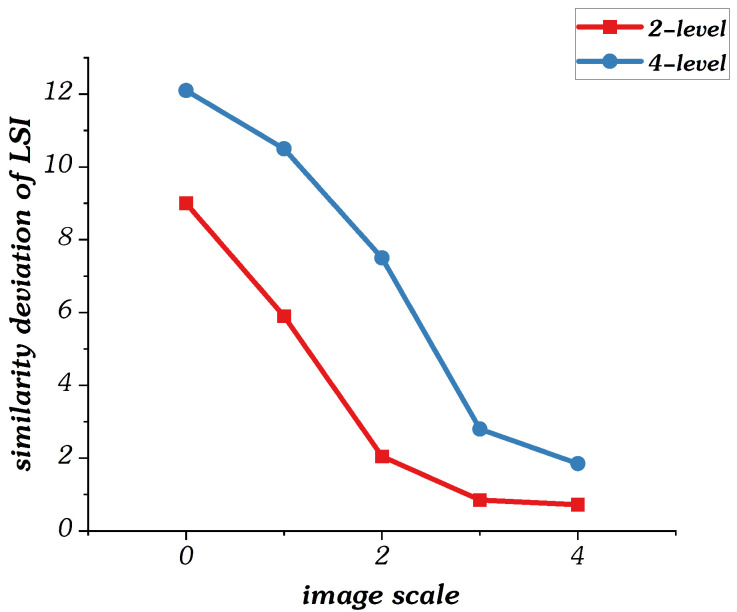
Curves of gradient similarity deviation values in multi-scale model.

**Figure 8 sensors-22-00499-f008:**
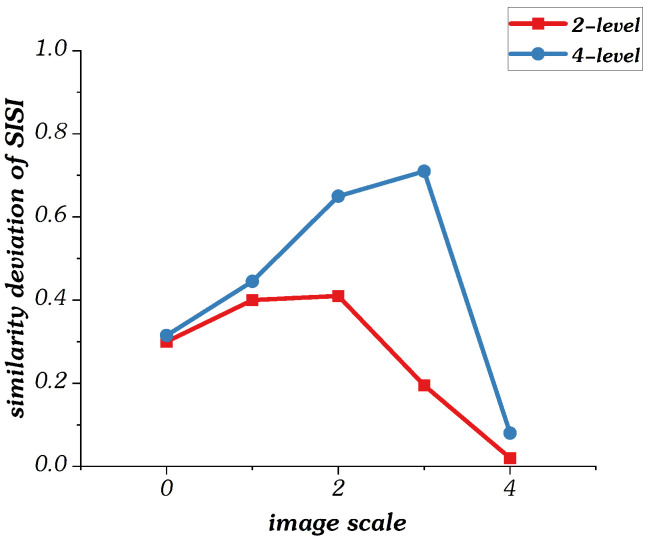
Curves of singular value similarity deviation values in multi-scale model.

**Figure 9 sensors-22-00499-f009:**
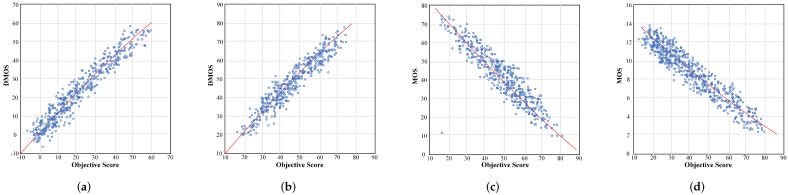
Scatter plots of overall predicted quality scores against the subjective scores of the proposed method on four Database. (**a**) LIVE Phase–I. (**b**) LIVE Phase–II. (**c**) WaterlooIVC Phase–I. (**d**) MCL.

**Table 1 sensors-22-00499-t001:** Performance of SIQA on four public databases, measuring the effect of cyclopean map and depth perception features on SIQA, respectively.

	${Q_{1}}_{\left(\mathrm{withcyclopeanmap}\right)}$			${Q_{1}}_{\left(\mathrm{withoutcyclopeanmap}\right)}$			$Q_{2}$			*Q*		
	PLCC	SROCC	RMSE	PLCC	SROCC	RMSE	PLCC	SROCC	RMSE	PLCC	SROCC	RMSE
LIVE Phase-I	0.9412	0.9278	5.2598	0.9389	0.9201	5.7892	0.8545	0.8458	8.5975	0.9512	0.9429	5.0028
LIVE Phase-II 3D	0.9325	0.9317	5.8256	0.9263	0.9136	5.2715	0.7789	0.7369	10.2548	0.9431	0.9452	4.2859
WaterlooIVC Phase-I	0.9458	0.9389	5.0214	0.9404	0.9321	5.5825	0.7782	0.7654	9.8975	0.9546	0.9478	4.2859
MCL	0.9124	0.9147	1.2925	0.9077	0.9101	1.2356	0.7625	0.7855	1.5478	0.9219	0.9259	1.0026

**Table 2 sensors-22-00499-t002:** Overall performance comparison of the proposed SIQA method and six methods on LIVE Phase-I and LIVE Phase-II 3D databases.

	LIVE Phase-I			LIVE Phase-II		
	PLCC	SROCC	RMSE	PLCC	SROCC	RMSE
Jiang [56]	0.9460	0.9378	5.3160	0.9261	0.9257	4.2627
Yue [45]	0.9370	0.9140	5.6521	0.9140	0.9060	4.4490
Khan [20]	0.9272	0.9163	-	0.9323	0.9272	-
Shao [57]	0.9389	0.9308	5.6459	0.9263	0.9282	4.1996
Geng [55]	0.9430	0.9320	5.5140	0.9210	0.9190	5.4001
Ma [54]	0.9409	0.9340	5.2110	0.9300	0.9218	**4.1232**
proposed	**0.9512**	**0.9429**	**5.0028**	**0.9431**	**0.9452**	4.2859

**Table 3 sensors-22-00499-t003:** Overall performance comparison of the proposed SIQA method and six methods on WaterlooIVC Phase-I 3D.

	PLCC	SROCC	RMSE
Khan [20]	0.9344	0.9253	-
Ma [54]	0.9252	0.9117	5.8766
Yue [45]	0.9261	0.9192	**4.6101**
Yang [49]	0.9439	0.9246	-
Geng [55]	0.8460	0.8101	9.4691
Proposed	**0.9546**	**0.9478**	4.6836

**Table 4 sensors-22-00499-t004:** Overall performance comparison of the proposed SIQA method and six methods on MCL 3D.

	PLCC	SROCC	RMSE
Zhou [53]	0.8850	0.8520	1.1770
Shao [58]	0.9138	0.9040	1.0233
Khan [20]	0.9113	0.9058	-
Liu [59]	0.9044	0.9087	1.1137
Chen [42]	0.8278	0.8300	1.4596
Proposed	**0.9219**	**0.9259**	**1.0026**

**Table 5 sensors-22-00499-t005:** In LIVE Phase-I and LIVE Phase-II 3D databases, performance comparison of the proposed SIQA method and six methods on different types of distortion, and the evaluation index is PLCC.

	LIVE Phase-I					LIVE Phase-II				
	JP2K	JPEG	Gblur	WN	FF	JP2K	JPEG	Gblur	WN	FF
Jiang [56]	0.9408	0.6975	0.9578	0.9516	0.8554	0.8463	0.8771	0.9845	0.9593	0.9601
Yue [45]	0.9350	0.7440	0.9710	0.9620	0.8540	**0.9860**	0.8430	0.9730	**0.9860**	0.9230
Khan [20]	0.9508	0.7110	0.9593	0.9470	0.8583	0.9270	0.8925	0.9778	0.9699	0.8987
Shao [57]	0.9366	0.6540	0.9542	0.9441	0.8304	0.8768	0.8506	0.9445	0.9339	0.9330
Geng [55]	0.9420	0.7190	0.9620	**0.9630**	0.8670	0.8510	0.8350	0.9790	0.9490	0.9480
Ma [54]	0.9610	0.7746	0.9711	0.9412	**0.8941**	0.9670	0.9350	0.9384	0.9341	0.9489
Proposed	**0.9679**	**0.7847**	**0.9787**	0.9558	0.8856	0.9327	**0.9452**	**0.9870**	0.9707	**0.9627**

**Table 6 sensors-22-00499-t006:** In LIVE Phase-I and LIVE Phase-II 3D databases, performance comparison of the proposed SIQA method and six methods on different types of distortion, and the evaluation index is SROCC.

	LIVE Phase-I					LIVE Phase-II				
	JP2K	JPEG	Gblur	WN	FF	JP2K	JPEG	Gblur	WN	FF
Jiang [56]	0.9027	0.6628	**0.9361**	0.9529	0.8079	0.8497	0.8547	0.9383	0.9563	**0.9555**
Yue [45]	0.8320	0.5950	0.8570	0.9320	0.7790	0.9590	0.7690	0.8680	**0.9590**	0.9130
Khan [20]	0.9074	0.6062	0.9295	0.9386	0.8092	0.9133	0.8670	0.8854	0.9584	0.8646
Shao [57]	0.9000	0.6339	0.9242	0.9430	0.7807	0.8747	0.8340	0.9241	0.9325	0.9409
Geng [55]	0.9050	0.6530	0.9310	**0.9560**	0.8160	0.8360	0.8410	0.9210	0.9390	0.9160
Ma [54]	0.9140	0.6659	0.9030	0.9037	**0.8312**	0.9328	0.8968	0.8992	0.8893	0.9167
Proposed	**0.9271**	**0.6758**	0.9252	0.9335	0.8156	**0.9636**	**0.9087**	**0.9398**	0.9319	0.9174

**Table 7 sensors-22-00499-t007:** In MCL 3D database, performance comparison of the proposed SIQA method and three methods on different types of distortion, and the evaluation index is PLCC.

	Zhou [53]	Shao [58]	Khan [20]	Liu [59]	Proposed
JPEG	0.8260	0.7016	**0.9574**	0.9404	0.9432
JP2K	0.8760	0.8571	0.9640	0.9219	**0.9725**
WN	0.9140	0.6748	0.9561	0.9135	**0.9345**
Gblur	0.9340	0.9013	0.9270	0.9479	**0.9603**
Sblur	0.9410	0.8640	0.9409	0.9530	**0.9600**
Tloss	**0.8910**	0.5814	0.8722	0.7618	0.8571

**Table 8 sensors-22-00499-t008:** In MCL 3D database, performance comparison of the proposed SIQA method and three methods on different types of distortion, and the evaluation index is SROCC.

	Zhou [53]	Shao [58]	Khan [20]	Liu [59]	Proposed
JPEG	0.7760	0.7992	0.8877	0.8506	**0.9045**
JP2K	0.8520	0.8415	0.9317	0.9011	**0.9320**
WN	0.9040	0.6404	**0.9517**	0.9256	0.9273
Gblur	0.9160	0.8993	0.9131	**0.9519**	0.9504
Sblur	0.9330	0.8532	0.9348	0.9577	**0.9617**
Tloss	0.8450	0.5674	0.8744	0.7909	**0.8818**

## Data Availability

The data presented in this study are available in the manuscript.

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
