# Peer review of "Rich Structural Index for Stereoscopic Image Quality Assessment"

_sensors, 2022, doi:10.3390/s22020499_

Round 1

Reviewer 1 Report

Major revision 
This manuscript introduces a Rich Structural Index (RSI) for Stereoscopic Image objective Quality Assessment (SIQA) method based on multiscale perception characteristics. author used Contrast Sensitivity Function (CSF), a locally adaptive manner on gradient maps which consider the luminance masking and contrast masking and Singular Value Decomposition (SVD) to get contrast index of reference and distortion component, finally they use Support Vector Machine Regression based on Genetic Algorithm (GA-SVR) training to get the final quality score. In summary, the research is interesting and provides valuable results, but the current document has several weaknesses that must be strengthened in order to obtain a documentary result that is equal to the value of the publication.
General considerations:
(1)The document contains a total of 66 employed references, of which 14 are publications produced in the last 5 years (21.21%), 23 in the last 5-10 years (34.85%), 29 than 10 years old (43.94%), implying a total percentage of 56.56 % recent references. In this way, the total number is sufficient, but their actuality is low, please cite more new papers.
(2)In this paper, it used 3 methods: LSI, SISI and DTSI. As for the fusion of the above three methods, please describe it in detail in the paper.
(3)Vision technology applications in various engineering fields, should also be introduced for a full glance of the scope of related area. The first paragraph introducing the research topic may present a much broad and comprehensive view of the problems related to your topic with citations to authority references:
Chen, M; Tang, Y; Zou, X; Huang, K; Li, L; He, Y. High-accuracy multi-camera reconstruction enhanced by adaptive point cloud correction algorithm. Optics and Lasers in Engineering 2019, 122, 170-183. https://doi.org/10.1016/j.optlaseng.2019.06.011
Cao, X.; Yan, H.; Huang, Z.; Ai, S.; Xu, Y.; Fu, R.; Zou, X. A Multi-Objective Particle Swarm Optimization for Trajectory Planning of Fruit Picking Manipulator. Agronomy 2021, 11, 2286. https://doi.org/10.3390/agronomy11112286
Wu, F.; Duan, J.; Chen, S.; Ye, Y.; Ai, P.; Yang, Z. Multi-Target Recognition of Bananas and Automatic Positioning for the Inflorescence Axis Cutting Point. Frontiers in Plant Science 2021, 12:705021. https://doi.org/ 10.3389/fpls.2021.705021
(4)In chapter 3, materials and methods, please specify the equipment used and describe the software and system in the experiment.
(5)Please indicate the physical or geometric meanings of specific parameters in Formulas 8-10.
(6)Please note that the graph is annotated in the text and corresponds to the real one. For example, in line 205, the specific algorithm description is shown in Figure 5.
(7)In lines 217-218, for the weight factors, please provide the artificial measurement standard or calculation evaluation formula in the paper.
(8)Please explain the specific criteria for 2-level and 4-level Gaussian blur.
(9)For the four public data sets selected, the size of samples is too small, please select a more appropriate data set.

Author Response

Thank you for your letter and the reviewers' comments on our manuscript entitled "rich structure index for stereo image quality assessment". These comments are very valuable and helpful for the modification and improvement of our paper, and also have important guiding significance for our researchers. We have carefully studied the comments and corrected them, hoping for approval.

Reviewer 2 Report

This paper presents a full-reference quality metric for stereoscopic images. The method proposed in this paper follows a well-known pipeline for designing learning-based quality metrics (i.e., feature extraction + aggregation + regression). This overall pipeline is widely used in the literature and the contributions of this paper are mostly focused on designing quality-aware features.

The paper is well written and a minor typo can be found on Page 6, line 189. There is a superscripted ".2" after the weight's equation.

In Section 3.4, the authors describe the use of GA-SVR. From the text, it is basically the ordinary SVR but with its hyperparameters optimized using GA. Why do authors use GA and not Random/Grid search? Why did the authors not use other regression methods such as random forest regression, XGBoost, etc? Remember that, if authors are doing hyperparameter optimization, the data should be split into training, validation, and testing. The 80/20 repeated random split should not be used in this case.

Still in Section 3.4, in its last paragraph, it is stated that experimental simulations were performed with 1000 repetitions of random 80/20 splits and the median was reported. This approach is quite common in similar literature. However, since the paper combines GA+SVR, I suspect that the model is overfitted. 

The overfitting risk is supported by the databases used in this work. The LIVE Phase1 has only 20 original stimuli. The other databases have even lesser pristine contains. I presume that the authors did not isolate the pristine/reference contents from the training/testing subset. In learning-based quality assessment, more than split in training/testing subset, it is crucial to segregate all occurrences of training content (reference + impaired) from the testing content. If this segregation is not performed, the correlation scores can be high but the method still does not work as a generic quality metric.

Since the benchmark datasets are small, my suggestion to authors is to adopt both grouped K-fold and leave one group out (LOGO) cross-validations (CV). In this case, the groups would be made according to the content/scene. For instance, in a given iteration of LOGO, only a single testing content should be selected from the original subset. All its processed/impaired correspondences should also be used for testing in this iteration. I hope this can help the authors:

  • https://scikit-learn.org/stable/modules/generated/sklearn.model_selection.GroupKFold.html
  • https://scikit-learn.org/stable/modules/generated/sklearn.model_selection.LeaveOneGroupOut.html

I also recommend cross-database validation. In this CV, a whole database is used for training and another database is used for testing. It is used to test the generalization of the model. If the proposed model is not overfitted and if it can be used for general purposes, the cross-database correlation should be also high as K-folded and LOGO CV.

Last but not least, I believe that authors should include a feature analysis with no regression. In problems like IQA, regression is used to fuse the features and to scale the outputs to a range next to the subjective scores. Therefore, an analysis with no regression is fundamental to demonstrate the method's effectiveness. 

Since the authors proposed a set of features, the readers will expect a correlation analysis using the raw features and the subjective scores. The authors can use a table where each row is a correlation index and each column is a single feature (correlation matrix). This analysis can also be performed using a scatter matrix. See: https://pandas.pydata.org/pandas-docs/stable/reference/api/pandas.plotting.scatter_matrix.html

Finally, I would like to clarify that this paper can be publishable, but these problems about data analysis must be resolved. We need more evidence that the metric works and the presented results look good due to the wrong regression analysis. 

Author Response

Thank you for your letter and for the reviewers' comments concerning our manuscript entitled "Rich Structural Index for Stereoscopic Image Quality Assessment". Those comments are all valuable and very helpful for revising and improving our paper, as well as the important guiding significance to our researcher. We have studied comments carefully and have made correction which we hope meet with approval.

Reviewer 3 Report

  • The chapters “Introduction” and “Related Works” need to summarize the challenges and limitations of current studies, rather than simply listing all the works of others. Because now it is difficult to understand the purpose of this study.
  • The manuscript is written in very poor English and is full of grammar and punctuation errors (even in abstract).
  • The article has many similarities to the manuscript "Evaluation of Stereoscopic Image Quality in the Context of Binocular Visual Mechanisms" in terms of writing style and structure, presentation of chart tables and parameters.
  • The measured value should be provided (for example, R-squared) not only graphical representation based on Scatter plots.
  • The GA-SVR model has been proposed by other authors, but references to their study are not included, and it appears that the authors have provided a new version of such an approach.
  • Penalty parameter c and the g2 should be explained
  • There are a variety of ways to perform GA selection, crossing, and mutation operations. None of them are explained;
  • The fitness function is not provided.

Author Response

(The authors gave the same response as above.)

Reviewer 4 Report

Your work, in my opinion, is a synthesis of known methods to achieve a new technical quality. It is advisable to re-edit the text to make it clear what benefits will be obtained by applying the algorithms proposed in the paper. I realize that this is only editorial cosmetics but it will improve the readability of the paper. Whether the data in the tables, could not be presented in a slightly different more readable form, the use of bold font alone is insufficient. The paper includes a very good literature review.

Author Response

(The authors gave the same response as above.)

Round 2

Reviewer 1 Report

ACCEPT

Reviewer 2 Report

The paper is publishable now.

Reviewer 3 Report

Thank you for the answers.